# Effects of Irrigating with Brackish Water on Soil Moisture, Soil Salinity, and the Agronomic Response of Winter Wheat in the Yellow River Delta

**Tianyu Wang**, **Zhenghe Xu * and Guibin Pang**

School of Water Conservancy and Environment, University of Jinan, Jinan 250022, China;
2015210161@mail.ujn.edu.cn (T.W.); stu_panggb@ujn.edu.cn (G.P.)

* Correspondence: stu_xuzh@ujn.edu.cn

**Abstract:** Water shortages due to low precipitation and seawater intrusion in the Lower China Yellow River Delta have occurred in recent years. Exploiting underground brackish water through well drilling is a potential alternative way to satisfy the demand for agricultural irrigation. However, how to successfully use brackish water for irrigation has become a new problem to solve. A two-year field experiment was conducted in this typical saline-alkaline region to investigate the effects of irrigating with brackish water on the soil water-salt dynamics, and the physiological response of winter wheat to drought-salt stress. The experiment was laid out in a randomized block design with three replications according to the quantity (160 mm and 240 mm) and quality (fresh water and brackish water with a salt concentration of 3 g L$^{-1}$) of irrigation water: T1 was 240 mm of fresh water, T2 was 160 mm of fresh water, T3 was 80 mm of fresh water and 160 mm of brackish water, and T4 was 80 mm of fresh water and 80 mm of brackish water. The results showed that the soil moisture of T3 was almost the same as T1 after the harvest of winter wheat each year, therefore, irrigating with brackish water can maintain soil moisture while saving fresh water resources. After two years, the soil salinity of each treatment increased by 0.307, 0.406, 0.383, and 0.889 g kg$^{-1}$, respectively. During the jointing-flowering stage, salt stress has a significant inhibitory effect on photosynthesis; T3 and T4 were lower than T1 and T2 in terms of plant height and dry weight. During the filling stage, because the effect of drought stress is more serious than that of salt stress, the photosynthesis of T3 was greater than that of T2 and T4. For both years, the yield of crops followed the rank order T1 > T3 > T2 > T4. Compared with irrigating with fresh water in T1, T3 changed the second and third irrigation into brackish water, however we did not find that soil salinity increased significantly, and this treatment was able to ensure crop growth during the filling stage. Therefore, the combination of fresh water (80 mm), then brackish water (80 mm), then brackish water (80 mm) is a feasible irrigation strategy in China's Yellow River Delta for winter wheat.

**Keywords:** irrigation strategies; water quality and quantity; soil salt accumulation; agronomic traits; photosynthesis

---

## 1. Introduction

The Yellow River Delta, located in eastern China's Shandong Province, is a vast territory that is abundant in natural resources. With the rapid development of agriculture in recent years, a reduction of water resources in this area has intensified. There is a shortage of surface water resources in the Yellow River Delta region and a serious imbalance between the supply and demand for water. The Yellow River is the main source of fresh water in this area and cannot meet the demands of local agricultural industries [1]. However, this area is rich in saline groundwater resources, with salinity ranging from

3 g L$^{-1}$ to 10 g L$^{-1}$ and a water table depth ranging from 2 m to 3 m, which makes the groundwater easy and inexpensive to extract [2]. Therefore, a careful use of the local subterranean brackish water could provide a new way to resolve the water resource shortage in this area.

Many countries and regions around the world that lack freshwater resources have been developing irrigation strategies using saline waters for a long time, and the related technologies are improving. In Italy, Hamdy et al. found that fresh water can be substituted with saline water to some extent, up to high values of Electrical Conductivity (EC) (9 dS m$^{-1}$), without any loss in production where there is deficit irrigation [3]. In US, Wiedenfeld found irrigating sugarcane with well water with an EC of 3.4 dS m$^{-1}$ resulted in a cane and sugar yield reduction of 17% from 82.9 to 68.8 Mg ha$^{-1}$ compared to the river water, with an EC of 1.3 dS m$^{-1}$ [4]. Mansour et al. used subsurface drip irrigation with saline water in Egypt, and found that soil salinity can be effectively controlled and has a preemptive effect on both wheat grain and straw yield [5]. In China, Liu et al. found that in the winter wheat-summer maize planting system, it is a feasible irrigation strategy to use brackish water to irrigate part of the crop growth period [6]. Ma et al.'s research also proved that irrigating with brackish water will not result in serious crop yield reduction [7]. However, brackish water irrigation also carries the risk of causing soil salinization and affecting the normal growth of crops. Nicolás et al. argued that irrigating with saline reclaimed water will affect crop yield and quality, so it is necessary to find a reasonable irrigation and soil management strategy when using brackish water [8]. Selim et al. proved that the salinity at the soil surface increased as inter-plant emitter distances and emitter depth increased [9]. Rahil et al.'s study recommended to use a short irrigation interval (one day interval) when highly saline water is used [10]. Tedeschi & Menenti also proved that irrigation frequency had an effect on soil salinity accumulation when using brackish water [11]. To reduce the damage to the soil associated with brackish water irrigation, many researches have concluded that adopting an effective irrigation system (such as drip irrigation, sprinkler irrigation) and strategy (such as high frequency irrigation) can control the degree of soil salinity accumulation and reduce the influence of high salinity on the soil environment. Moreover, brackish water can alter the soil environment [12] and affect the soil capillary action, resulting in changes to the soil permeability and water retention [13]. It can also lead to the accumulation of salts in the soil, which will restrict crop growth and cause physiological drought [14,15], affecting water uptake by crop roots, inhibiting photosynthesis, and altering the physiological characteristics of crops. Therefore, the effect of brackish water on soil and crops should be fully considered, when using brackish water for irrigation.

In China, some studies have proved that irrigation with brackish water is feasible in the North China Plain [16], in the eastern coastal areas of China; some areas of saline soil caused by sea water intrusion have also been experimented with by using brackish water for irrigation [17]. In the Yellow River Delta, there are few studies on the use of brackish water to irrigate winter wheat. Flood irrigation is currently the main irrigation system used by winter wheat farmers, and in order to avoid soil salinization when irrigating with brackish water, water quality and amount should be carefully controlled. In this study, we used a field experiment to (1) explore the effects of brackish irrigation on the physicochemical properties of the soil and the agronomic response of winter wheat, and (2) find a feasible irrigation strategy for applying brackish water in the Yellow River Delta, to control soil salinization and ensure crop growth.

## 2. Materials and Methods

### 2.1. Experimental Site

We conducted field experiments in the city of Zhanhua in Shandong Province (37°34′ N, 117°45′ E, Figure 1). The local climate is monsoon at mid-range latitudes. During the period of 1979 to 2000, the annual average sunshine was about 2686 hours and the annual average temperature was 12.9 °C. The average annual precipitation was 549.8 mm and the rainy season was concentrated from June to August. With an evaporation–precipitation ratio of 3.22, moisture is lost easily by evaporation. Figure 2

shows the temperature and precipitation dynamics from October 2015 to June 2017 in Zhanhua; there was no precipitation during the irrigation period in both years in the experimentation area. The Yellow River Delta groundwater depth is 1–3 m; because of overexploitation in recent years, the groundwater table has decreased significantly, causing seawater intrusion, and the degree of soil salinization is serious [18]. Through the well in the experimentation area, the groundwater depth is 3–4 m. The soil characteristics before the experiment are reported in Table 1; loam and sandy loam are the main soil texture in our experimental area, the average soil salinity at 0–60 cm is 1.93 g kg$^{-1}$, and the average pH is 7.17, with mildly salinized soil.

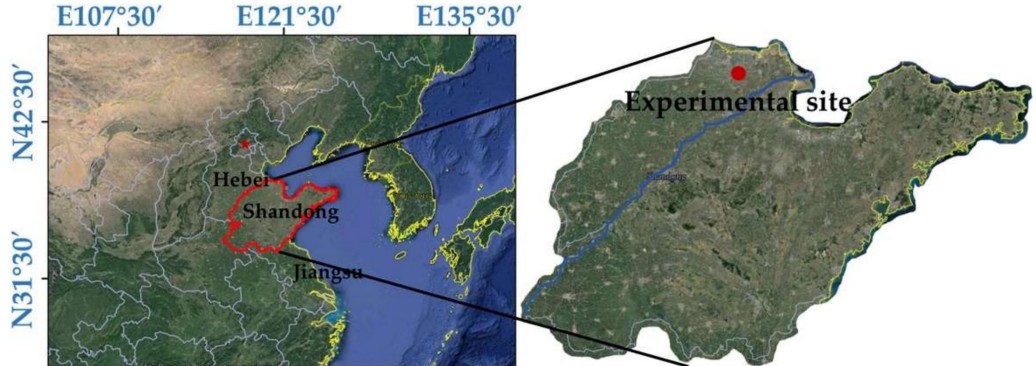

**Figure 1.** Location of the experimental site. The field experiments were carried out in Zhanhua City, Shandong Province, located in the lower reaches of the Yellow River Delta.

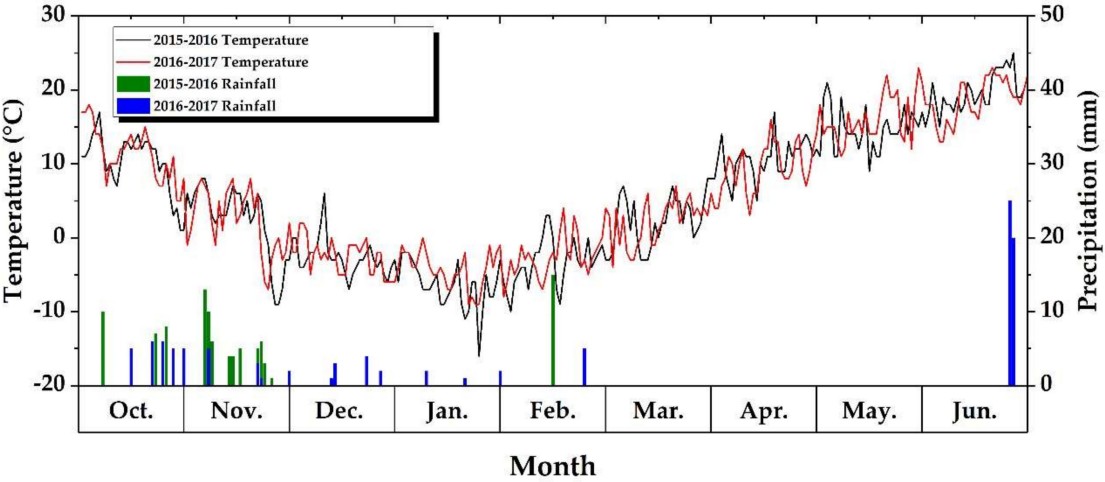

**Figure 2.** Temperature (°C) and precipitation (mm) during winter wheat growth periods (2015.10–2017.05).

**Table 1.** Physical and chemical characteristics of the soil, including bulk density, pH, ion concentration, total salt content, and soil texture.

| Soil Layer (cm) | Bulk Density (g·cm$^{-3}$) | pH | CO$_3^{2-}$ (g·kg$^{-1}$) | HCO$_3^-$ (g·kg$^{-1}$) | Cl$^-$ (g·kg$^{-1}$) | SO$_4^{2-}$ (g·kg$^{-1}$) | Ca$^{2+}$ (g·kg$^{-1}$) | Mg$^{2+}$ (g·kg$^{-1}$) | K$^+$ (g·kg$^{-1}$) | Na$^+$ (g·kg$^{-1}$) | Total Salt (g·kg$^{-1}$) | Soil Texture |
|---|---|---|---|---|---|---|---|---|---|---|---|---|
| 0~20 | 1.39 | 7.30 | 0.01 | 0.01 | 0.51 | 0.46 | 0.10 | 0.03 | 0.10 | 0.14 | 1.36 | loam |
| 20~40 | 1.33 | 7.13 | 0.01 | 0.02 | 1.95 | 0.42 | 0.12 | 0.02 | 0.08 | 0.24 | 2.86 | sandy loam |
| 40~60 | 1.32 | 7.07 | 0.01 | 0.02 | 0.76 | 0.36 | 0.07 | 0.02 | 0.08 | 0.23 | 1.56 | sandy loam |
| 60~80 | 1.36 | 7.03 | 0.01 | 0.01 | 0.25 | 0.18 | 0.10 | 0.03 | 0.07 | 0.12 | 0.78 | loamy sand |
| 80~100 | 1.46 | 7.03 | 0.00 | 0.01 | 0.27 | 0.22 | 0.14 | 0.02 | 0.06 | 0.14 | 0.85 | loam |

## 2.2. Experimental Design

A field experiment was carried out during the winter wheat growing seasons for 2015–2017, using a randomized block design with three replicates. The area of each block was 28 m$^2$ (4 m × 7 m). Before sowing, we embedded plastic waterproof film to a depth of 1.5 m in the whole experiment site and between each plot to prevent any interaction (Figure 3). Winter wheat has a shallow root and low salt tolerance during its recovery stage [19], so we used fresh water in the four treatment areas for the first irrigation (2016.3.12/2017.3.16), then we established the four treatments, combining two levels of irrigation water quantity (160 mm and 240 mm) and quality (freshwater with salt concentration less than 0.5 g L$^{-1}$ and brackish water with a salt concentration of 3 g L$^{-1}$) (Table 2). A brackish water desalination unit was used to desalinate the groundwater to the required salt content. A local winter wheat cultivar, Jimai 22#, was planted in 14 rows in each plot, with a plant spacing of 20 cm. Following local planting practices, we applied 10 kg of fermented chicken manure to each plot before sowing. We then added 750 g of urea fertilizer to each plot during the recovery stage. Other conditions (such as ploughing and loosening the soil before sowing, regular weeding, and pesticide spraying) were consistent with conventional practices.

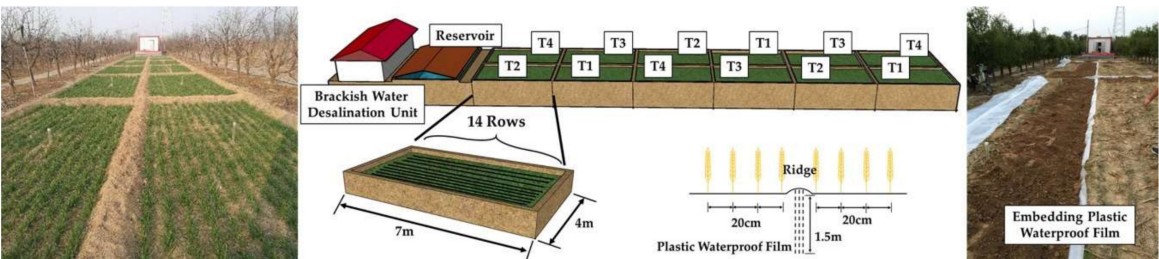

**Figure 3.** Experimental design used in the current study.

**Table 2.** Description of the irrigation strategy (including irrigation water quantity and quality), and each important date (including the date of sowing and harvesting, and the date of each irrigation), and the length of the growing cycle.

| | Irrigation Quota and Irrigation Time | | | Growing Cycle | | |
|---|---|---|---|---|---|---|
| | First Irrigation mm | Second Irrigation mm | Third Irrigation mm | Sowing | Harvest | Cycle |
| T1 | 80 freshwater | 80 freshwater | 80 freshwater | 2015.10.08 | 2016.06.13 | 249 days |
| T2 | 80 freshwater | 80 freshwater | 0 | | | |
| T3 | 80 freshwater | 80 brackish water | 80 brackish water | 2016.10.08 | 2017.06.12 | 248 days |
| T4 | 80 freshwater | 80 brackish water | 0 | | | |
| Date | 2016.3.11/2017.3.15 | 2016.4.27/2017.4.25 | 2016.5.22/2017.5.14 | | | |

The first irrigation was conducted at the recovery stage, the second at the jointing–flowering stage, and the third at the filling stage.

## 2.3. Experimental Methods

### 2.3.1. Soil Sampling and Analysis

Soil moisture content: A Pico-BT Mobile Moisture Measurement Trime Meter (TRIME-PICO IPH/T3) (EIC, Germany) was used to measure the volumetric soil water content in each plot at depths of 0–20 cm, 20–40 cm, 40–60 cm, 60–80 cm, and 80–100 cm. We performed measurements at irrigation and at the sowing, recovery, jointing, filling, and maturity stages of winter wheat. At every stage, each replication was measured once, and three measurements were taken for each treatment.

Soil salinity: Soil samples were taken from depths of 0–20 cm, 20–40 cm, 40–60 cm, 60–80 cm, and 80–100 cm with a drill in each plot on the same dates described for the soil moisture content,

sampling three times per treatment. We extracted the soil at a soil: water ratio of 1:5, and used a DDS-307 conductivity meter (INESA Scientific Instrument, China) to measure the EC. We randomly selected 30 samples of soil and used the drying method to determine the soil salinity; based on the EC values of these 30 samples the formula for converting electrical conductivity to soil salinity was fitted (Figure 4), then we translated the EC of all samples into soil salinity by using the equation $y = 0.00261x - 0.05113$.

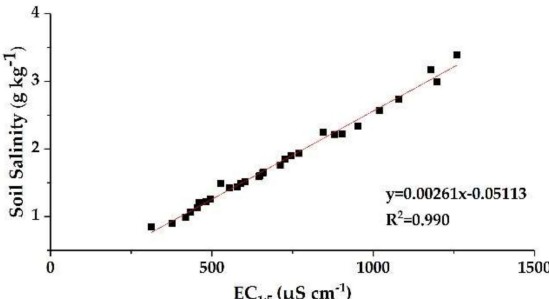

**Figure 4.** Relationship between the electrical conductivity of 1:5 soil:water extract ($EC_{1:5}$) and soil salinity.

### 2.3.2. Crop Physiological Indicators

The photosynthetic rate ($P_n$) and transpiration rate ($T_r$) were measured using an LCpro-SD portable photosynthetic apparatus (ADC, UK). During the jointing–flowering and filling stages, three fully expanded flag leaves were selected from each plot and measured every 2 hours from 8:00 to 18:00 on sunny days, once every two days in a week.

At each growth stage of winter wheat, we selected five winter wheat plants from each plot during every stage and measured their heights.

We randomly selected a row of wheat from each plot and sampled the whole plant in the range of 10 cm*20 cm. Then, the leaf areas were measured using a CI203 (CID, USA). The ratio of the total leaf area to the sample area was taken as the Leaf Area Index (LAI).

At each growth stage of winter wheat, we selected 10 representative plants from each plot and dried them at 105 °C for 30 min to stop internal crop reactions, then the dry weight was obtained after drying samples at 70 °C for 48 h.

After harvest, in each plot, we selected 1 $m^2$ winter wheat as the sample, measured the 1000 grain weights, 1 $m^2$ winter wheat yield, 1 $m^2$ spike number, and the grain number per spike. Then, the remaining winter wheat yield in the plot was recorded, and the output of the sample was added as the final yield of each plot, which was converted into the yield per hectare.

### *2.4. Data Analysis*

Data were analyzed using an analysis of variance (ANOVA; statistical software IBM SPSS 19) to determine differences between treatments. Differences among mean values were calculated using the least significance difference (LSD) at the 5% level.

## 3. Results

### *3.1. Soil Moisture Content*

We found the soil water content at 0–40 cm fluctuated greatly during the winter wheat growth period, and increased sharply after each irrigation (Figure 5). With increasing depth, the soil is less affected by temperature, evaporation, rainfall, and irrigation, so the soil water content changes slowly. After the second irrigation, at the depth of 0–20 cm, soil moisture content of T3 and T4 were higher than that of T1 and T2. In other words, the soil moisture content following irrigation with brackish water was higher than following irrigation with fresh water. In addition, we found that the soil water

content of the T3 treatment was higher than that of T1 under the same irrigation quantity at a depth of 0–20 cm in both years. This phenomenon was also seen at depth of 20–60 cm in 2017. When the experiment ended in 2017, the final soil moisture contents of T1 and T3 ranged from 10.3% to 19.2% at a depth of 0–100 cm, and those of T2 and T4 ranged from 5.9% to 15.8%.

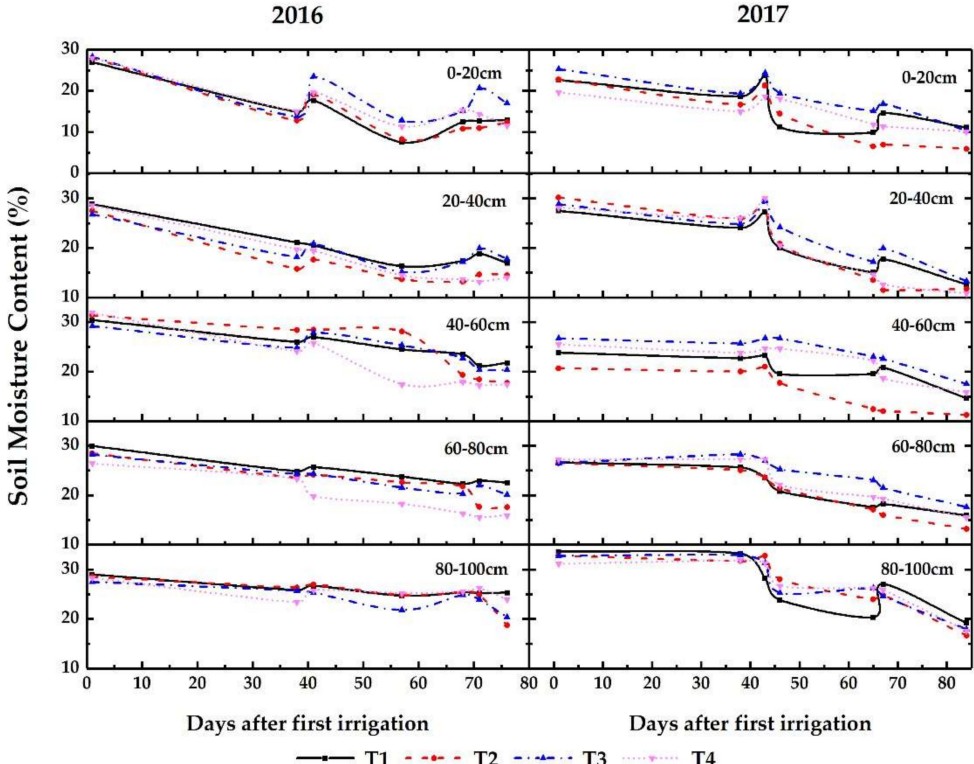

**Figure 5.** Soil moisture content of four treatments after first irrigation at a depth of 0–100 cm. T1 represents irrigation of 240 mm fresh water, T2 represents irrigation of 160 mm fresh water, T3 represents irrigation of 80 mm fresh water and 160 mm brackish water with a salt concentration of 3 g L$^{-1}$, and T4 represents irrigation of 80 mm fresh water and 80 mm brackish water with a salt concentration of 3 g L$^{-1}$.

*3.2. Soil Salinity*

Figure 6 shows the changes in soil salinity over the two year study period. The distribution law of soil salinity contents for all the treatments was basically the same before planting the winter wheat (2015.10.1), and salt accumulated mainly at the depth of 20–40 cm. The soil salinities at 0–20 cm and 20–40 cm were 1.39 g kg$^{-1}$ and 2.79 g kg$^{-1}$, respectively, indicating salinization.

At first irrigation, soil salinity was effectively controlled by irrigating with fresh water at this stage. At the jointing–flowering stage (2016.4.20/2017.4.22), there was a slight accumulation of soil salinity contents at a depth of 0–40 cm. After the second irrigation (2016.4.24/2017.4.30), soil salinity in T1 and T2 decreased, whereas it increased in T3 and T4 at a depth of 0–40 cm due to the brackish water. The average salinity at 0–40 cm, in 2016, increased under T3 from 1.13 g kg$^{-1}$ to 2.33 g kg$^{-1}$ and under T4 from 1.09 g kg$^{-1}$ to 2.28 g kg$^{-1}$; in 2017, T3 increased the salinity from 1.08 g kg$^{-1}$ to 1.30 g kg$^{-1}$ and T4 from 1.09 g kg$^{-1}$ to 2.05 g kg$^{-1}$. At the filling stage (2016.5.14/2017.5.11), soil salinization was aggravated due to the enhanced evaporation associated with high temperatures. After the third irrigation (2016.5.21/2017.5.19), the salinity of T1 in each soil layer was below 1.1 g kg$^{-1}$ in both 2016 and 2017. The salinity of the soil surface in T3 increased from 2.52 g kg$^{-1}$ to 2.67 g kg$^{-1}$ in 2016, and from 1.82 g kg$^{-1}$ to 2.06 g kg$^{-1}$ in 2017; however, the salinity of the deep soil decreased due to water leaching. Soil salinization was aggravated in T2 and T4; the high salt content of T2 in 2016 was related to the lack of irrigation and the high temperature and evaporation at that time.

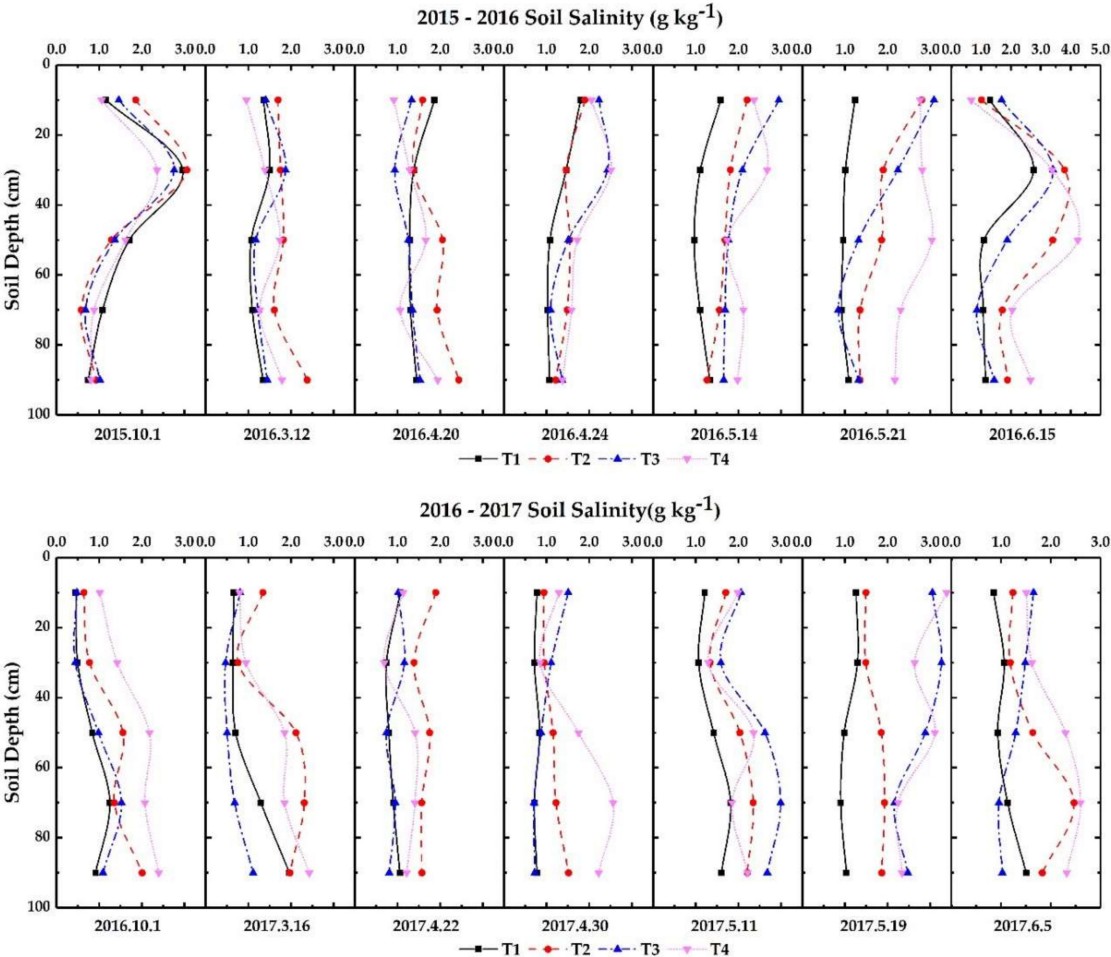

**Figure 6.** Soil salinity of four treatments at a depth of 0–100 cm in two growing cycles. T1 represents irrigation of 240 mm fresh water, T2 represents irrigation of 160 mm fresh water, T3 represents irrigation of 80 mm fresh water and 160 mm brackish water with a salt concentration of 3 g $L^{-1}$, and T4 represents irrigation of 80 mm fresh water and 80 mm brackish water with a salt concentration of 3 g $L^{-1}$. (**a**): 2015–2016; (**b**): 2016–2017.

Table 3 shows the soil salinity fluctuations after a growth period. During the 2015–2016 growth period, we used fresh water in T1 for all three irrigations and found the soil salinity decreased. The other three treatment areas showed varying degrees of salt accumulation in the following order: T3 < T2 < T4. During the 2016–2017 growth period, salt accumulation occurred in all four treatment areas, following the same order of T1 < T3 < T2 < T4, as observed during the previous period. T2 and T4 were only irrigated twice, and the final soil salinity was higher than that of T3, which used brackish water for the third irrigation.

**Table 3.** Changes of soil salinity during growth period. T1 represents irrigation of 240 mm fresh water, T2 represents irrigation of 160 mm fresh water, T3 represents irrigation of 80 mm fresh water and 160 mm brackish water with a salt concentration of 3 g L$^{-1}$, and T4 represents irrigation of 80 mm fresh water and 80 mm brackish water with a salt concentration of 3 g L$^{-1}$.

| Year | Treatment | Initial Total Salinity (g kg$^{-1}$) | Final Total Salinity (g kg$^{-1}$) | Variation in Soil Salinity (g kg$^{-1}$) |
|---|---|---|---|---|
| 2015–2016 | T1 | 1.530 | 1.477 | −0.053 |
| | T2 | 1.544 | 2.362 | 0.818 |
| | T3 | 1.458 | 1.854 | 0.396 |
| | T4 | 1.346 | 2.599 | 1.253 |
| 2016–2017 | T1 | 0.792 | 1.099 | 0.307 |
| | T2 | 1.268 | 1.674 | 0.406 |
| | T3 | 0.902 | 1.285 | 0.383 |
| | T4 | 1.182 | 2.071 | 0.889 |

### 3.3. Physiological Response of Wheat Plants

Since the blade area of the wheat plant is so small at the recovery stage, photosynthesis was not measured. The winter wheat was thriving in this stage, and although the four treatment areas showed some differences in their plant heights and LAI, no differences were observed for dry weight (Figure 7).

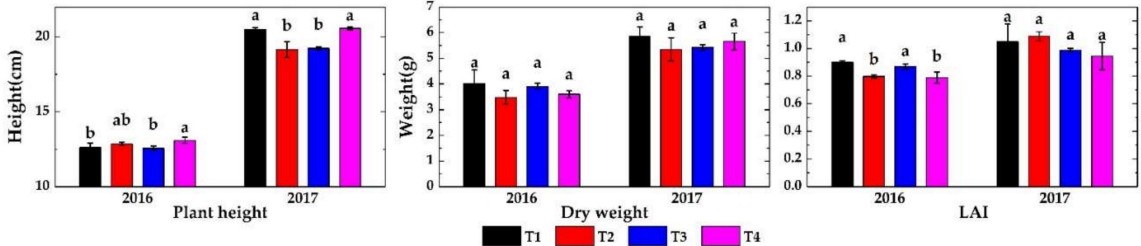

**Figure 7.** Physiological indices during the recovery stage. T1 represents irrigation of 240 mm fresh water, T2 represents irrigation of 160 mm fresh water, T3 represents irrigation of 80 mm fresh water and 160 mm brackish water with a salt concentration of 3 g L$^{-1}$, and T4 represents irrigation of 80 mm fresh water and 80 mm brackish water with a salt concentration of 3 g L$^{-1}$. Bars with the same letter are not significantly different according to the least significance difference test at $p \leq 0.05$. Error bars are standard errors. (**a**): Plant height; (**b**): Dry weight; (**c**): LAI.

In 2016, the photosynthesis of the wheat plants at the jointing–flowering stage showed the same situation in the four treatment areas before and after the second irrigation (Figure 6a,c,e). The differences in the four treatment areas before irrigation were also small in 2017 (Figure 6b), but due to the increased soil salinity, the photosynthesis values of T3 and T4 were lower than those of T1 and T2 after irrigation (Figure 6). Although the soil moisture content can be replenished by brackish water, the soil salinity increases rapidly at the same time. High salinity can then lead to a physiological drought and reduce the crop water uptake, thereby resulting in the inhibition of photosynthesis (Figure 8).

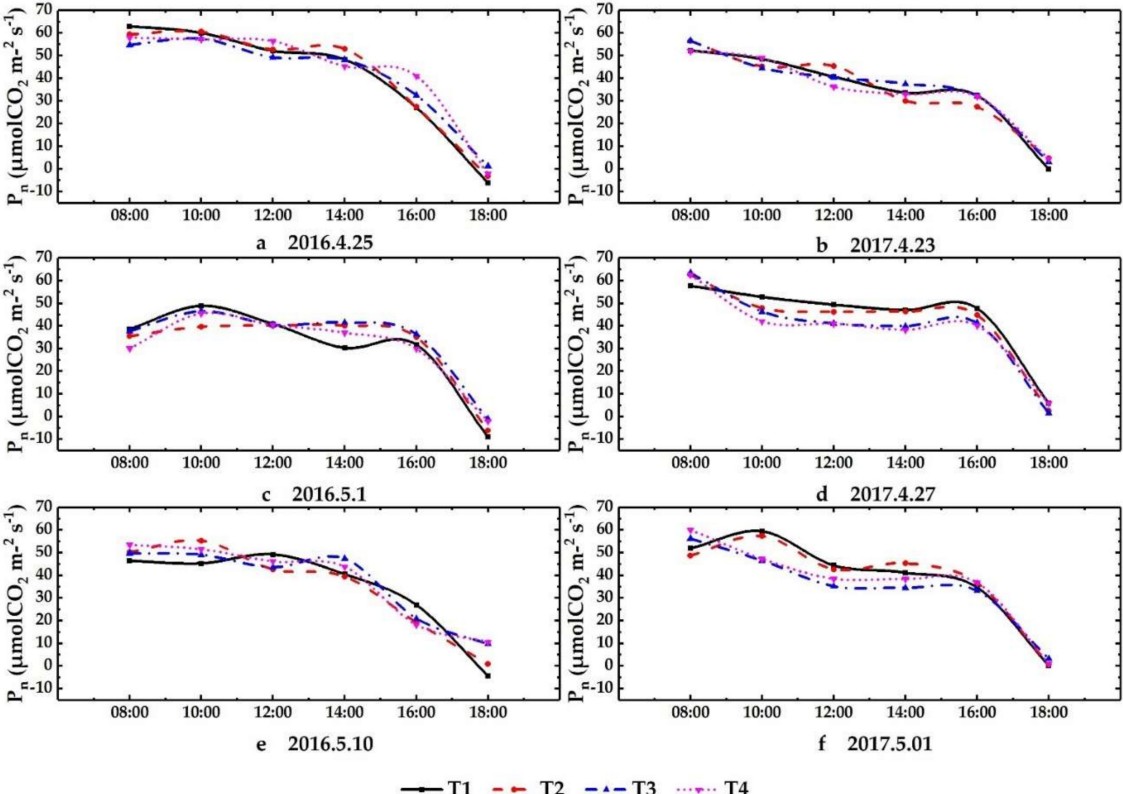

**Figure 8.** Daily evolution of the photosynthesis rate during the jointing–flowering stage. T1 represents irrigation of 240 mm fresh water, T2 represents irrigation of 160 mm fresh water, T3 represents irrigation of 80 mm fresh water and 160 mm brackish water with a salt concentration of 3 g L$^{-1}$, and T4 represents irrigation of 80 mm fresh water and 80 mm brackish water with a salt concentration of 3 g L$^{-1}$. (**a**): 2016.4.25; (**b**): 2017.4.23; (**c**): 2016.5.1; (**d**): 2017.4.27; (**e**): 2016.5.10; (**f**): 2017.5.01.

The growth rate of winter wheat was highest during this stage, and the crop morphologies were diverse under salt stress. T3 and T4 showed significant decreases in plant height and dry weight in comparison with T1 and T2. The average plant height of T3 and T4 was 7.7% lower than that of T1 and T2 in 2016, and 7.3% in 2017. The average dry weight of T3 and T4 was 16.2% lower than that of T1 and T2 in 2016, and 6.7% in 2017. In contrast, we observed no significant differences in the LAI of wheat among the different treatments (Figure 9).

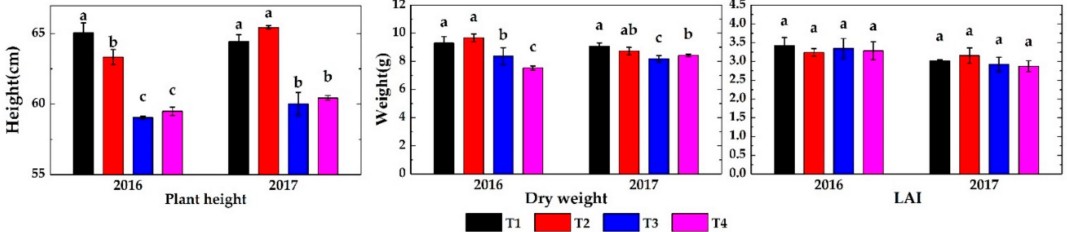

**Figure 9.** Physiological indexes during the jointing–flowering stage. T1 represents irrigation of 240 mm fresh water, T2 represents irrigation of 160 mm fresh water, T3 represents irrigation of 80 mm fresh water and 160 mm brackish water with a salt concentration of 3 g L$^{-1}$, and T4 represents irrigation of 80 mm fresh water and 80 mm brackish water with a salt concentration of 3 g L$^{-1}$. Bars with the same letter are not significantly different according to the least significance difference test at $p \leq 0.05$. Error bars are standard errors. (**a**): Plant height; (**b**): Dry weight; (**c**): LAI.

As shown in Figure 8, in 2016, there were no differences in the photosynthesis of plants from different treatments during the filling stage before and after the third irrigation (Figure 10a,c,e). Before the third irrigation, the photosynthesis of all treatments was basically the same in the daytime, and at 16:00, the photosynthesis of the T3 treatment was greater than that of other treatments (Figure 10b), but when we supplied T1 with fresh water, and supplied T3 with brackish water, the plant photosynthesis in these treatments was higher in both than that in T2 and T4. Despite the increased salinity of the surface soil, the photosynthesis of T3 was still greater than that of T2 and T4 (Figure 10d,f).

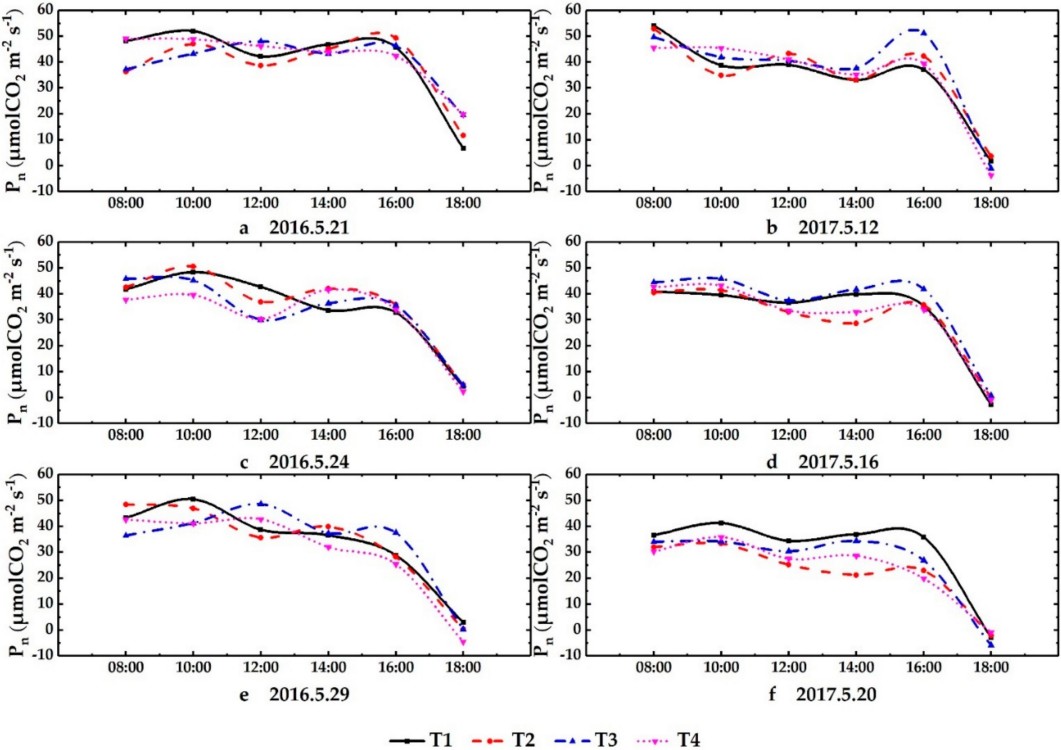

**Figure 10.** Daily evolution of the photosynthesis rate during the filling stage. T1 represents irrigation of 240 mm fresh water, T2 represents irrigation of 160 mm fresh water, T3 represents irrigation of 80 mm fresh water and 160 mm brackish water with a salt concentration of 3 g $L^{-1}$, and T4 represents irrigation of 80 mm fresh water and 80 mm brackish water with a salt concentration of 3 g $L^{-1}$. (**a**): 2016.5.21; (**b**): 2017.5.12; (**c**): 2016.5.24; (**d**): 2017.5.16; (**e**): 2016.5.29; (**f**): 2017.5.20.

The growth rate of winter wheat slowed down upon entering the filling stage, and we observed some similarities in the heights and dry weights of the wheat plants over the two years: the plant height of T1 was highest and that of T4 was lowest, and the dry weights of T2 and T4 were significantly lower than those of T1 and T3. When the winter wheat entered the filling stage, the leaves began to shrink, so the leaf area index of this stage decreased slightly. We observed no significant differences in the different treatment areas in 2016. In 2017, however, T2 experienced a significant decrease in its LAI compared with that of T3 and T4 (Figure 11).

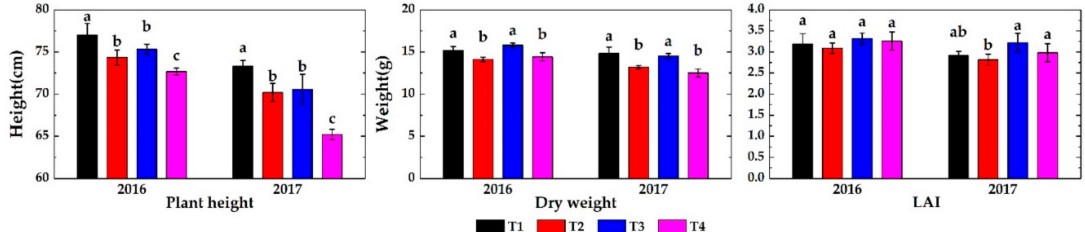

**Figure 11.** Physiological indexes during the filling stage. T1 represents irrigation of 240 mm fresh water, T2 represents irrigation of 160 mm fresh water, T3 represents irrigation of 80 mm fresh water and 160 mm brackish water with a salt concentration of 3 g $L^{-1}$, and T4 represents irrigation of 80 mm fresh water and 80 mm brackish water with a salt concentration of 3 g $L^{-1}$. Bars with the same letter are not significantly different according to the least significance difference test at $p \leq 0.05$. Error bars are standard errors. (**a**): Plant height; (**b**): Dry weight; (**c**): LAI.

### 3.4. WUE and Winter Wheat Yield

Table 4 lists the winter wheat final harvest situation. For both years, the plant height followed the order T1 > T3 > T2 > T4, and the heaviest dry weight was observed in T1, followed by T3, then T2 and T4. There was no significant difference in 1000 grain weight among treatments. In 2016, when comparing treatments irrigated with the same quality, the yield of T2 was 18.1% lower than that of T1, and the yield of T4 was 14.0% lower than that of T3; when comparing treatments irrigated with equal quantities of water, the yield of T3 was 5.9% lower than that of T1 and the yield of T4 was 1.1% lower than that of T2. In 2017, when comparing treatments irrigated with the same quality, the yield of T2 was 17.9% lower than that of T1, and the yield of T4 was 8.7% lower than that of T3; when comparing treatments irrigated with the same quantities of water, the yield of T3 was 10.6% lower than that of T1 and the yield of T4 was 0.5% lower than that of T2. For both years, the WUE of T1 and T3 was significantly lower than that of T2 and T4, because there was a small difference in yield among treatments, but there was a significant difference (160 mm/240 mm) in irrigation water quantity. The yields of the four treatment areas in 2017 were reduced by 8.1%, 0.80%, 12.7%, and 7.4%, respectively, compared with those of 2016.

**Table 4.** Winter wheat harvest situation over two years. T1 represents irrigation of 240 mm fresh water, T2 represents irrigation of 160 mm fresh water, T3 represents irrigation of 80 mm fresh water and 160 mm brackish water with a salt concentration of 3 g $L^{-1}$, and T4 represents irrigation of 80 mm fresh water and 80 mm brackish water with a salt concentration of 3 g $L^{-1}$.

| Year | Treatment | Plant Height (cm) | Dry Weight (g) | 1000-Grain Weight (g) | Block Yield (kg) | WUE (kg m$^{-3}$) | Crop Yield (kg ha$^{-1}$) |
|---|---|---|---|---|---|---|---|
| 2015–2016 | T1 | 77.67 ± 0.18 a | 32.37 ± 0.81 a | 45.89 ± 1.09 ns | 19.76 ± 0.93 a | 4.57 ± 0.22 b | 7055.95 ± 333.12 |
| | T2 | 74.33 ± 0.08 c | 26.38 ± 0.20 c | 44.98 ± 1.43 ns | 16.19 ± 0.99 b | 5.62 ± 0.34 a | 5782.14 ± 351.91 |
| | T3 | 76.09 ± 0.56 b | 29.42 ± 0.73 b | 45.80 ± 2.47 ns | 18.60 ± 1.81 a | 4.31 ± 0.42 b | 6642.86 ± 645.62 |
| | T4 | 73.33 ± 0.75 d | 27.65 ± 1.32 c | 44.99 ± 1.27 ns | 16.00 ± 1.09 b | 5.56 ± 0.38 a | 5715.47 ± 645.62 |
| 2016–2017 | T1 | 76.23 ± 0.90 a | 30.68 ± 0.77 a | 41.22 ± 1.37 ns | 18.15 ± 0.19 a | 4.20 ± 0.04 b | 6683.33 ± 66.27 |
| | T2 | 72.61 ± 0.64 b | 27.35 ± 0.13 b | 38.73 ± 1.34 ns | 14.90 ± 1.20 b | 5.17 ± 0.42 a | 5321.43 ± 426.89 |
| | T3 | 73.75 ± 0.37 b | 30.52 ± 0.61 a | 40.13 ± 1.30 ns | 16.24 ± 1.27 ab | 3.76 ± 0.29 b | 5798.81 ± 453.51 |
| | T4 | 70.37 ± 0.49 c | 26.33 ± 0.45 c | 37.91 ± 2.49 ns | 14.82 ± 1.66 b | 5.15 ± 0.58 a | 5294.05 ± 593.66 |
| *P*-Value | | | | | | | |
| Irrigation quantity | | <0.01 | <0.01 | >0.05 | <0.01 | <0.01 | \ |
| Irrigation quality | | <0.05 | <0.05 | >0.05 | >0.05 | >0.05 | \ |
| Irrigation quantity × Irrigation quality | | <0.01 | <0.01 | >0.05 | < 0.01 | <0.01 | \ |

Values within each column followed by the same letter(s) are not significantly different according to the least significant difference test ($p \leq 0.05$).

## 4. Discussion

After one year of the experiment, we found that even the irrigation strategy of T3 (fresh water then brackish water then brackish water) could supplement soil moisture and reduce salt accumulation [20]. After two years, we found that with the same irrigation quantity, brackish water irrigation can ensure soil moisture content. In our experiment, the surface soil moisture content of irrigation with brackish water was even higher than that of irrigation with fresh water, possibly because brackish irrigation changed the physical and chemical properties of the soil [21,22], or it may be that brackish water reduces the absorption of water by crops, which is in accordance with the research results of Yuan et al. [23]. However, according to the research and classification of Katerji et al. [24], wheat belongs to the salt-tolerant and drought-tolerant group, which has a high threshold value for salinity and drought tolerance. The brackish water with 3g $L^{-1}$ is still within the tolerance of crops and can be used for irrigation. On the premise of meeting the water requirements of winter wheat, we saved the fresh water resource by utilizing the abundant underground saltwater resources.

When using brackish water for irrigation, the salt accumulation in soil should be considered. If the cumulative salinity is too high over the long term, the soil environment and crops will be affected. Alban Echchelh et al. evaluated the feasibility of using marginal water for irrigation. They thought that soil texture was one of the main factors affecting soil salinity, and as the percentage of sand increases, the leaching of soil salt was also more obvious [25]. The soil in our experimental area was mainly sandy loam, which will reduce the adhesion of salt in the soil so that the salt can be better leached with water. According to Feng et al. [26], the desalination efficiency of soil decreases with the increase of salt in irrigation water. Liu et al. analyzed the feasibility of long-term brackish water irrigation for winter wheat-summer maize. They believed that irrigation time was the main factor affecting soil salt accumulation characteristics. Under the condition of irrigation once a year, soil salt accumulation was basically stable [16]. In our experiment, brackish water was used for irrigation once or twice during the winter wheat growing period. According to the changes of soil salinity shown in Table 3, soil salt accumulation in T4 was the highest over the two years, and T3 had a smaller salt accumulation. Some studies have shown that there is a close relationship between groundwater depth and soil salinity [27,28]: when the depth of groundwater is shallow and the salt content of groundwater is high, soil salinization will be aggravated due to evaporation. Our results show that the brackish water with a salt concentration of 3 g $L^{-1}$ still has some leaching effect on soil salinity in our experimental area, which can control soil salinization to a certain extent. Therefore, if the quality and quantity of the brackish water is reasonable, soil salinity can also be controlled. Moreover, the groundwater table may decline slightly by pumping for irrigation; this pumping also reduced the potential for soil salinization due to shallow groundwater. After two consecutive years of irrigation, the soil salinity was still acceptable. After the winter wheat harvest, the Yellow River Delta enters its rainy season, and rainfall further leaches out the soil salt; this will alleviate the effect of brackish water irrigation on soil salinization. However, in dry years, the irrigation method should be adjusted to the use of fresh water to prevent excessive salt accumulation, and the effects of brackish water irrigation on the soil should also be continuously monitored.

Recent studies have reported that gene selection, exogenous hormone regulation, and other methods can be used to improve the salt tolerance of crops [29,30]. Farooq et al. [31] summarized the effects of salt stress on crops and stated that nutrient management may improve salt tolerance in grain legumes. Amin et al. [32] showed that adding potassium and zinc to brine irrigation water could reduce the toxic effect of salinity on wheat. However, it is difficult to implement these methods under the management of farmers, so the regulation of irrigation remains a feasible and effective way for reducing salt stress. In this experiment, we simulated an ordinary planting field undergoing brackish water irrigation. Winter wheat cultivars are commonly grown locally and have not been optimized to improve their salt tolerance. The amount of fertilizer applied is also intended to ensure the normal growth of crops without increasing the amount of fertilizer applied to alleviate salt stress on crops.

At the same time, irrigation water also comes from local groundwater, and the irrigation strategy is similar to local planting habits.

The duration and intensity of stress will affect the photosynthesis of crops [33,34]. In our experiment, we observed the change of photosynthesis rates of winter wheat before and after irrigation. Brackish water can cause some salt stress, but it can also relieve drought stress. After irrigation in jointing–flowering stage, there was no obvious difference in photosynthesis between treatments in 2016 (F8ce), and there was a slightly inhibited photosynthesis in T3 and T4 in 2017 (F8df). As time went by, the photosynthesis of each treatment basically returned to the same situation at the start of the filling stage (F10ab), and the influence of brackish water irrigation declined gradually over time. After irrigation in the filling stage, the photosynthesis of T3 underwent a slight increase at 10:00 am (F10d). According to Ors and Suarez [35], moderate salt stress may increase crop yields and promote plant growth. In our experimental area, the most common irrigation strategy is to irrigate winter wheat two or three times during the growing season, with an irrigation interval of 30 to 40 days. Although salt stress is caused by using brackish water for irrigation, the effect of water on the crops is still very obvious within 3–5 days after irrigation. As a salt-tolerant crop, winter wheat does not suffer a serious decline in photosynthesis under the condition of irrigating with brackish water with a salt concentration of 3 g $L^{-1}$.

When penetrating the soil, salt is absorbed by roots and spread to leaves, affecting the growth of crops to different degrees [36]. Moreover, the effects of brackish water on crops are different at different growth stages [37]. The jointing stage is the fastest changing stage of winter wheat height, which is closely related to the final plant height of winter wheat. According to Liu's research, using brackish water to irrigate in the jointing period for winter wheat is a good option [6]. In our experiment, the plant heights and dry matters of T3 and T4 were lower than those of T1 and T2 during the jointing stage. Until the filling period, under the dual stress of water and salt, T4 plants were shorter and had lower dry matter contents, while T3 plants were less affected. Irrigation with brackish water once or twice during the growth period of winter wheat could lower the final plant heights of the crop, but in north China where windy weather appears frequently in May and June, Wang's research shows that excessive plant height is one of the causes of winter wheat lodging [38], and lower plant height can improve the lodging-resistance capability of the crop. In addition, the T4 irrigation strategy inhibited dry matter accumulation in crops. In contrast, the dry matter accumulation of T3 was almost the same as that of T1. We thought that the salt present in the brackish water may result in a physiological drought, thereby inhibiting water absorption, reducing plant height and dry matter content, however the effect of irrigation water amount on crop agronomic traits is more obvious than that of water quality.

The variation of LAI can reflect crop stress [39,40]. Our experiment indicates that winter wheat LAI is more sensitive to water stress than to salt stress. If brackish water irrigation is adopted in the jointing period, salt stress may not cause a large decline of LAI. In the filling period, insufficient irrigation may accelerate leaf aging. In this period, a fast decline of LAI may affect photosynthesis and filling of the crop, and reduce the final yield. In our experiments, we found that the LAI of crops was less affected by salt stress, and there was no obvious leaf atrophy even under stress.

As in the study by Jiang et al. [41], the yield of winter wheat declined with the increase of salt in the irrigation water. Jiang's experiment simulated four uses of irrigation. Our experiment used an irrigation frequency of three times to adapt to the agricultural production habit of the experimental area in the Yellow River Delta, where farmers usually irrigate two or three times during the growing period of winter wheat. Zhang et al. [42] indicated that water stress affects the filling time of winter wheat, reducing yield. In our experiment, we set three irrigation modes in the filling period: fresh water (T1), brackish water (T3), and no irrigation (T2, T4). Although brackish water irrigation produced salt stress in winter wheat, the final yield of T3 was higher than those of T2 and T4, which did not receive water during the filling stage. The filling stage is particularly important to crop yield, and the effect of drought stress is greater than that of salt stress during this stage. Without irrigation in this stage, the crop yield will be reduced, which has also been reported by Chauhan et al. [43]. The effect

of salt stress is less serious than that of drought during the filling stage, and we found the lack of a third irrigation will suppress crop filling and, thus, the final yield. All the treatment areas in 2017 experienced a yield reduction compared to 2016, although the field experiment results were affected by many factors, including rainfall, temperature, evaporation, and so on. While this resulted in the yield fluctuations, we cannot ignore the effect of continuous use of brackish water in irrigation on crops. Therefore, further studies are needed to elucidate the effects that brackish water may have for the sustainability of crop yields and the maintenance of soil quality in the long-term.

## 5. Conclusions

Our results show that brackish water can be used to irrigate winter wheat in the Yellow River delta region, and the combination of 80 mm fresh water followed by 80 mm brackish water (salt concentration of 3 g L$^{-1}$) then followed by 80 mm brackish water (salt concentration of 3 g L$^{-1}$) is a reasonable and feasible irrigation strategy under the existing irrigation method of three irrigation times per growth period. With this combination, the soil moisture and the salt accumulation are maintained—second only to the fresh water–fresh water–fresh water irrigation strategy. In each growth period of winter wheat, brackish water with a salt concentration of 3 g L$^{-1}$ did not lead to a significant decline in photosynthesis of the crops. Under the condition of a long irrigation interval, a certain amount of brackish water irrigation could even promote photosynthesis of crops. During the jointing–flowering stage, salt stress can lead to physiological drought, which can affect plant height and dry matter accumulation, but a lower plant height is not undesirable, as it can improve the lodging-resistance capability of crops. During the filling stage drought stress is more harmful than salt stress, and even when brackish water is used for irrigation during the filling period, the final yield is still higher than that of the treatment without irrigation during this period. However, further monitoring of soil, crop growth and yield is still needed to make feasible long-term irrigation with brackish water.

**Author Contributions:** Conceptualization, Z.X. and G.P.; Formal analysis, T.W.; Funding acquisition, Z.X. and G.P.; Investigation, T.W. and G.P.; Writing (original draft), T.W.; Writing (review & editing), Z.X.

**Funding:** This research was funded by the National Natural Science Foundation of China (No.51509105) and Shandong Provincial Natural Science Foundation, China (ZR2014EEQ020). We are grateful to the anonymous reviewers and editor for their insightful comments and suggestions on this paper.

**Conflicts of Interest:** The authors declare no conflict of interest.

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
