# Peer review of "Effects of Irrigating with Brackish Water on Soil Moisture, Soil Salinity, and the Agronomic Response of Winter Wheat in the Yellow River Delta"

_sustainability, doi:10.3390/su11205801_

Round 1

Reviewer 1 Report

The manuscript entitled “Effects of Irrigating with Brackish Water on Soil Moisture, Soil Salinity, and Agronomic Response of Winter Wheat in Yellow River Delta” (Reference number Sustainability-595628-v1), authored by T. Wang, Z. Xu and G. Pang represents a great improvement from a previous submission to this journal, which I had the opportunity to review. I appreciate the effort made by the authors and thank them for taking into account my comments and suggestions. Furthermore, I congratulate the authors for the huge effort they put in improving their manuscript.

English has been greatly enhanced and the manuscript achieves the standard to be published in an international journal. Moreover, authors improved the description of the Materials and Methods, clarifying several aspects that were not clear.

Therefore, I recommend a minor revision of this manuscript prior to its acceptance for publication in Sustainability.

Specific comments to the authors:

Abstract:

Reduce the abstract since it is too long. According to the journal’s guidelines, the maximum number of words is limited to 200 and you surpassed this limit by far.

Introduction:

Greatly improved from the previous versions.

By the end of the second paragraph, use “when using brackish water for irrigation” instead of “when using brackish water irrigation”.

Materials and Methods:

Greatly improved from the previous submission.

When describing the measurement of soil moisture you stated “Each treatment was measured three times”. What do you mean? That you made three measurements per treatment or that you measured soil moisture on three dates per treatment? Did you measure soil moisture on each replication or only on one replication per treatment? Please, specify this.

Results:

Table 3: Please, in the third column, use “Variation in soil salinity” instead of “Change amount soil salinity”. Besides, check the superscript in the units for this column.

Figures 7: Check the significances in this figure. It seems that plant height did not vary much among treatments and you signalled significant differences. It is weird, especially when you look at the variability in your data (standard errors are overlapping among treatments). Please, check also the text on this issue.

Figure 11: Same comments as for figure 7.

Discussion:

This section improved from the previous version of the manuscript. However, there is still some obscure passage.

In the first paragraph, you stated: “brackish water inhibits the absorption of water by crops, which is the same as the research results of Yuan et al. [23]”. If this is correct, why using brackish water for irrigation when crops cannot absorbe it?

Later in this paragraph, you stated: “We think the reason for this result is related to the high evaporation and high salinity of groundwater in this area; brackish water with a salt concentration of 3 g L-1 is still has some leaching effect on soil salinity. Therefore, in our experimental area, salt concentration in irrigation water was not the only factor affecting soil salinity. If the quality and quantity of the brackish water is reasonable, soil salinity can be controlled. Moreover, we lowered the groundwater table by pumping before winter wheat planting, which also reduced the potential for soil salinization due to shallow groundwater”. This is confusing and must be re-phrased. I suggest reducing this, as well.

The idea in the next paragraph, “In this experiment, we simulated an ordinary planting field undergoing brackish water irrigation” must be developed further.

Later, you said: “We thought that the salt present in the brackish water may result in a physiological drought, thereby inhibiting water absorption, reducing plant height and dry matter content, however the effect of irrigation water amount on crop agronomic traits is more obvious than that of water quality”. This implies negative effects of using brackish water for irrigation, so you must explain further the benefits that using this water may bring instead of leaving crops rain-fed.

The last sentence of the discussion should be completed with “Therefore, further studies are needed to elucidate the effects that brackish water may have for the sustainability of crop yields and the maintenance of soil quality in the long-term”.

Conclusions:

From my viewpoint, this section is too long (20 lines) and can be reduced.

References:

Check the reference list very carefully because I detected that you used the names of the authors instead of the surnames. Besides, the references are not edited according to the journal guidelines.

Maybe you can use the following references for improving your introduction and discussion sections, since they are highly related to your study:

Letey, J.; Hoffmann, G.J.; Hopmans, J.W.; Grattan, S.R.; Suarez, D.; Corwin, D.L.; Oster, J.D.; Wu, L.; Amrhein, C. Evaluation of soil salinity leaching requirement guidelines. Agric. Water Manage. 2011, 98, 502-506; doi: 10.1016/j.agwat.2010.08.009

Mirás-Avalos, J.M.; Intrigliolo, D.S. Grape composition under abiotic constrains: Water stress and salinity. Front. Plant Sci. 2017, 8, 851; doi: 10.3389/fpls.2017.00851

Reviewer 2 Report

The authors made changes but at the Conclusions, they were not too inspired:

- repeat what was specified in the section 31 Experimental site („The groundwater in this area is shallow and the soil texture is mainly sandy loam.”);

- confused expressions are made („...brackish water irrigation still has a certain leaching effect on soil salinity…”);

- express themselves well-known things, as their own discoveries („…long-term use of brackish water irrigation may increase the risk of soil salinization…”).

- it is repeated that the best variant from the experienced ones is the irrigation with fresh water followed by two waterings with brackish water.

I think this part of the paper needs to be rewritten.

Reviewer 3 Report

The article was corrected and deserves publication

Author Response

Thank you very much for your affirmation of our work. At the same time, thank you again for your help. The comments you provided in the past helped us to improve our manuscripts.

This manuscript is a resubmission of an earlier submission. The following is a list of the peer review reports and author responses from that submission.

Round 1

Reviewer 1 Report

Line 29. In keywords repeat title words.

Line 69. The writing must be done impersonally. Example avoid sentences like "We done ...".

Line 74. Does it refer to the average temperature?.

Line 87. Perhaps in Table 1 it is convenient to also show data of electrical conductivity

             and the Ca, Mg, SO4 data?.

Line 90. It should be written 4 m x 7m.

Line 99. It would be desirable to express the salinity also in dS/m.

Line 142. Soil moisture is expressed on the basis of mass or volume?.

Line 249. Indicate WUE units

                Expand the discussion of WUE results based on crop evapotranspiration data

Author Response

Thank you very much for your valuable comments on the manuscript. We revised the manuscript according to your comments, and improve our writing. According to the editor's suggestion, I used the "Track Changes" to modify the manuscript, and I also responded to your comments use the Microsoft Office Word's "Comments" function in our manuscript, you can review these contents through the manuscript I submit.

We appreciate for your warm work earnestly, and hope that the correction will meet with approval. If you still have questions about our manuscript, we are still will to continue to revise our manuscript.

Once again, thank you very much for your comments and suggestions,

Reviewer 2 Report

I appreciate the Editor to give the chance to review an interesting and valuable paper. I found merits in the both methodology and results.

The article is devoted to the analysis of the brackish water irrigation effects on soil properties and on the physiological response of plants to the drought-salt stress.

Based on the study, the authors identify pertinent solutions to satisfy the demand of agricultural irrigation in a safe and effective way.

Also, the manuscript is written in an agreeable style.

However, I recommend to the authors to experience at least one vegetation season.Two years of experimentation is a time too short to draw conclusions fair in this field.

Author Response

Thank you very much for your comments on our manuscript and for giving us the opportunity to continue to revise it. We have made a lot of changes in the manuscript, and you can review these contents through the manuscript I submit.

About your comment for the time of experiment, we have got similar results through our two-year experiment. There was no obvious contradiction between the two years experiment. Therefore, it is feasible to draw some conclusions based on the two-year experiment results.

We appreciate for your warm work earnestly and did our best to revise the manuscript, and hope that the correction will meet with approval. If you still have questions about our manuscript, we are still will to continue to revise our manuscript.

Once again, thank you very much for your comments and suggestions.

Reviewer 3 Report

The manuscript entitled “Effects of brackish water irrigation on soil water-salt and physiological response of winter wheat in arid areas” (Reference number sustainability-495543), authored by T. Wang, Z. Xu and G. Pang presents results from a two-year study in which authors compared the response of wheat to four different irrigation treatments (two doses and two water qualities combined), as well as on the soil water and salt contents. Authors found that salt content in the soil was not significantly increased when irrigating with brackish water, although crop yield was slightly diminished. They concluded that combining fresh water with brackish water would be a means to save fresh water without compromising wheat yields. The experiment seems to be carried out with a high degree of care and the results are interesting. However, I am not sure that this manuscript belongs within the scope of the journal Sustainability because it seems more appropriate to a journal devoted to agriculture or irrigation management (such as Agriculture, Agronomy or Water, all of them from MDPI). I leave to the editor to decide on this.

Anyway, the manuscript must be greatly improved because of the English language (which is repetitive and misleads readers with many confusing portions) and the presentation of the results, which is rather poor.

A second concern that I have deals with the discussion, which does not deal with the sustainability of the practice recommended by authors (irrigation with brackish water). Moreover, this section is very weak and repeats results all over.

The introduction can also be improved because authors cited a lot of previous works but do not provide a summary of the main results about the topic they are studying. Therefore, readers are not informed about the state of the art, and the objective is not well introduced.

Finally, I suggest authors to be more cautious with their conclusions and the extent of these conclusions.

In the following lines, I provide the authors with a great number of suggestions in order to improve their work. Therefore, I recommend the rejection of this manuscript because it does not fit within the scope of the journal, as it is written. However, if editor decides to give the authors an opportunity to modify their manuscript, the suggestions in the following pages must be considered so the manuscript achieves the high-quality standards for being published in Sustainability.

Specific comments to the authors:

Title:

As it is written it does not make sense and readers would think that you worked on several areas with an arid climate, which is not true. Therefore, I suggest substituting it to “Effects of irrigating with brackish water on soil water and salt contents and on the agronomic response of winter wheat in an arid area”.

Abstract:

The abstract is informative and fairly written. However, it has to be improved because it is long and I detected some contradictions.

Line 9: “has been starved of fresh water”, I do not see what you mean.

Line 10: “develop”, did you “develop” the brackish water? How?

Line 12: “brackish irrigation” must be changed to “irrigating with brackish water”.

Line 13: “dynamics” instead of “dynamic”.

Line 15 and line 17 (in many other portions of the text): “3 g L-1 brackish water”, what is this? It is repeated all over the manuscript and it is clearly incorrect. It should be, if I understood well, “brackish water with a salt concentration of 3 g L-1”.

Line 19: Remove “content”.

Lines 21-22: “brackish water irrigation does not significantly increase soil salinity”, not true looking to what it happened to T4.

Lines 25-26: “indicating brackish water irrigation can promote photosynthesis at this stage”, not true, it was the amount of water that caused this effect, not its quality.

Line 26: Remove “the two”.

Lines 27-28: Remove this sentence. You never explained how water was applied, so this conclusion does not make sense when reading the abstract.

Keywords:

Line 29: “winter wheat, brackish irrigation” must be removed since they already appear in the title.

Introduction:

This section is very brief (which is not bad) but does not focus the readers on the objective of the manuscript. Authors should use better the references that they consulted, and maybe use some new ones in order to set correctly the problem to study and its implications on the environmental and agricultural sustainability, which is the main topic of the journal in which they intend to publish their work.

Lines 33-34: Remove “is the youngest delta in China. This”.

Line 35: “reduction of water resources in this area is intensified” instead of “water resources problems in this area have gradually intensified”.

Line 37: “demand for water. The Yellow River is the mainly source” instead of “demand for water resources. The Yellow’s River water, as the mainly source”.

Lines 38-39: “However” instead of “On the other hand”.

Line 39: Check units, correct superscripts.

Lines 39-42: Citations are needed here.

Line 41: Include “a” before “careful”. Remove “resources”.

Line 42: Remove “resources”.

Lines 43-56: In this paragraph, you cited a lot of works but did not explain the results obtained in any of them.

Line 43: Remove “sufficient”.

Line 44: “irrigation strategies using saline waters” instead of “brackish water irrigation”.

Line 49: “strategy” instead of “irrigation method”. What do you mean by “sub-soil irrigation”?

Line 53: “salts” instead of “salt”. “which will restrict crop growth” instead of “which will stress the growth of crops”.

Lines 54-55: Remove “Salinity is an important exogenous stress factor in plant growth [18], which”.

Line 55: “affecting water uptake by crop roots” instead of “affects the moisture absorption of the crop root system”.

Line 56: What do you mean by “ecological” here?

Line 57: “Border irrigation”? What is this? Use “system” instead of “method” and remove “ordinary”.

Line 58: “which causes soil salinization in arid and semi-arid regions”. Untrue, it depends on water quality.

Lines 58-59: “By choosing...” Re-phrase this sentence.

Lines 59-61: Not true, there are many examples of research focusing on final crop yield as affected by the use of salty water for irrigation. Even you cited some of them.

Line 62: “field planting scale”, what is this?

Line 63: “physiological ecology”, what?

Lines 64-66: This must be re-phrased.

Materials and Methods:

The experimental design and sampling seem to be carried out with a high degree of care. This gives strength to this study. However, this section needs further information to fully describe the experiment which has been carried out.

Lines 71-73: Indicate the years for which these averages are calculated. Are they 30-year averages?

Line 74: “dynamics” instead of “statistics”.

Line 75: Remove “according to field records”. “was” instead of “is”. “in both years” instead of “both two years”.

Line 76: Remove “in the experimentation area”.

Line 78: “and the degree of soil salinization is serious”, that of soil or that of groundwater... or both?

Lines 79-80: Separate “m” from “3-4”. Remove “in the experimental area”.

Lines 80-81: This sentence should be re-phrased to “Soil characteristics before the experiment are reported in Table 1”.

Figure 1a and Figure 1b: Either you use two different figures or join them into a single one with two panels. The captions should be joined as well. Besides, these captions must stand alone and you do not need to include abbreviations (such as T for temperature or R for rainfall) when you do not use them in the graphs.

Table 1: Data shown in this table must be briefly commented in the text. Besides, you do not need to include a slash (/) between the name of the variable and its units. “Soil texture” instead of “Soil type”. Use two decimals for all numbers within the table.

Line 89: “seasons” instead of “growth periods”.

Line 91: “sowing” instead of “seeding”.

Line 92: “treatments” instead of “treatment”.

Line 95: Include “with a plant” before “spacing”.

Line 96: Remove “seeds”.

Lines 97-98: Be clearer about these “other conditions”.

Line 99: The title of this table should be “Description of the irrigation treatments established and dates of the wheat growing cycle over the two seasons studied”.

Table 2: You do not need to include a slash (/) between the name of the variable and its units (which must be given between parentheses). Remember to use “brackish water with a salt concentration of 3 g L-1”. In the “Cycle” include “(days)” instead of using “d” after the numbers. In the footnote, use “the second one” and “the third one” instead of “the second irrigation” and “the third irrigation”, respectively.

Figure 2: The caption should be re-phrased. A suggestion: “Experimental design used in the current study”.

In general, when stating the model of an apparatus, provide the name of the brand and the country between parentheses.

Line 106: Remove “Soil moisture content:”

Lines 106-107: Re-phrase to “Before sowing, a probe for measuring soil water content (Pico-BT Mobile Moisture Measurement Trime Meter) was installed in each plot at depths of”.

Lines 108-109: Why not continuous measurements with the probe? Was the same probe used in all plots? In that case you did not bury the probe but some tubes for introducing it into the soil. This description must be clarified.

Line 110: “Soil samples were taken” instead of “Soil salinity: In the field, we drilled soil samples”.

Line 111: Remove “respectively. We performed these measurements at” and use simply “on”.

Line 112: “dates” instead of “time points”.

Line 113: With a conductivity meter you measure electrical conductivity but not “salinity” and much less salt content, which is shown in this manuscript. Therefore, this methodology must be clarified.

Line 114: “Crops” instead of “Crops”.

Line 115: Use subscripts for “n” and “r” in “Pn” and “Tr”. “were measured every 10 days” should be before “after recovery stage” and “Pn and Tr” can be removed.

Line 118: Remove “portable photosynthetic apparatus manufactured by” and use “ADC” between parentheses.

Line 121: How the same lighting conditions? In fact, lighting conditions change in 10 hours.

Line 122: Remove “Plant height:”

Line 124: Remove “Leaf area index (LAI):” What do you mean by “sampled 10 cm”?

Line 126: It should be “200 cm2”. “we measured their leaf areas using a CI203 area measuring instrument”, repetitive. Re-phrase it.

Line 128: Remove “Dry weight:”

Line 129: “processed them at 105 ºC for 30 min”, what do you mean? “their constant weight”, this is funny. It is easier to say “dry weight was obtained after drying samples at 70 ºC for 48 hours”.

Line 131: Remove “Crop yield:”

Line 132: “and” before “grain number per spike”.

Line 133: “recorded” instead of “counted”.

Line 134: “yield per hectare” instead of “yield of hectares”.

Lines 136-137: This first sentence can be removed.

Line 140: “and LSD was performed using IBM SPSS 19”, only the LSD or also the ANOVA?

Results:

This section presents some inconsistencies between the text and the information shown in the tables. Besides, English is rather bad and should be greatly improved.

Line 143: Re-phrase to “Soil water content at 0-40 cm fluctuated greatly”.

Line 144: “and increased” instead of “and to increase”.

Lines 144-145: “Both fresh and brackish water can be used to replenish soil moisture”, it is better to say that you did not find any difference in water content between irrigation water qualities.

Lines 147-149: Depending on depth? On year? Please, clarify.

Line 151: The caption of this figure does not stand alone. It must be re-phrased.

Lines 153-154: Remove “in which we can see that the”.

Line 154: “consistent”, with what?

Line 155: Check superscripts in the units.

Line 156: “indicating salinization” instead of “which are slightly salinized”.

Lines 156-158: Remove the last sentence of this paragraph.

Lines 159-160: This sentence belongs to Materials and Methods.

Line 161: Remove “the” before “soil salinity”.

Line 162: “soil salinity contents were stable” instead of “the soil salinity was stable”. “with only a slight accumulation”, where?

Line 163: “soil salinity in T1 and T2 decreased by” instead of “the soil salinities of T1 and T2 were further decreased by”.

Line 164: “whereas increased in T3 and T4 at a depth of 0-40 cm due to” instead of “whereas the soil salinities of T3 and T4 at a depth of 0-40 cm had increased due to”.

Line 165: Remove “irrigation” and check the superscripts in the units.

Lines 166-167: Check the superscripts in the units. For what depth are you talking about?

Line 170: Remove “We irrigated T3 with brackish water, and found”. Use “the salinity of soil surface in T3 increased” instead of “the salinity of its surface soil to increase”.

Line 171: Check superscripts in the units.

Line 172: Use a semicolon before “however” and remove “had”. Remove “We did not irrigate T2 and T4, so we found the”.

Line 173: “was aggravated in T2 and T4” instead of “to be even further aggravated”.

Line 175: “occurred” instead of “occurs”. “reducing” instead of “which reduces”.

Line 176: Remove “area began to enter the” and “which”.

Figure 4: How could you obtain salt contents if you measured electrical conductivity? Separate “(cm)” from “Depth” in the Y-axis.

Line 179: The caption for this figure does not stand alone and it must be re-phrased.

Line 182: Remove “1.recovery Stage: For the first irrigation, we used fresh water in the four treatment areas”.

Line 183: “so small at recovery stage, photosynthesis was not measured” instead of “so small at this stage, we did not measure its photosynthesis”.

Line 185: “in their plant heights and LAI, whereas no differences were observed for dry weight” instead of “in their plant heights and leaf area indexes, we found no significant differences in dry weight”.

Figure 5: You can substitute this figure for a table in order to reduce space.

Lines 188-189: This caption does not stand alone. Authors must indicate that the figure shows the variables for each treatment and year.

Line 190: Remove “2. Jointing-Flowering Stage:” and use “at the jointing-flowering stage” after “wheat plants”.

Line 191: Remove “basically”.

Lines 192-193: “This means that the brackish water...” this sentence can be removed.

Line 197: “crop water uptake” instead of “the crop moisture absorption”.

Figure 6: Use the same scale for the Y-axis in all graphs. Besides, what are you representing? You never defined “A” and this is not photosynthesis nor transpiration. Moreover, the units are wrongly expressed because you never used slashes (/) before, but superscripts.

Line 200: This caption does not stand alone and must be re-phrased.

Line 206: “indexes of wheat from the different treatments” instead of “indexes of the different treatment areas”.

Figure 7: You can substitute this figure for a table in order to reduce space.

Lines 208-209: This caption does not stand alone. Authors must indicate that the figure shows the variables for each treatment and year.

Line 210: Remove “3. Filling Stage:”

Line 211: “of plants from the different treatments during the filling stage before and after” instead of “of the different treatment areas before and after”.

Line 213: “we supplied T1 and T3 with brackish water”, you never used brackish water for T1.

Lines 214-215: “higher” instead of “stronger”.

Lines 215-218: “This is because...”, this sentence must be removed since it is not results. You could move it to the discussion section if you consider it useful.

Figure 8: Use the same scale for the Y-axis in all graphs. Besides, what are you representing? You never defined “A” and this is not photosynthesis nor transpiration. Moreover, the units are wrongly expressed because you never used slashes (/) before, but superscripts.

Line 220: This caption does not stand alone and must be re-phrased.

Line 223: “lower” instead of “less”.

Lines 224-226: “The leaf area index...” These sentences can be removed.

Lines 228-229: “and the values of T1 and T3 were higher than those of T2 and T4”, not true according to the figure.

Figure 9: You can substitute this figure for a table in order to reduce space.

Lines 231-232: This caption does not stand alone. Authors must indicate that the figure shows the variables for each treatment and year.

Line 233: Why capital letters for “Wheat”?

Line 235: “the dry weight of T1 was heaviest and that of T4 was lightest”, this is not well written.

Line 236: Include “comparing treatments” after “When”.

Lines 237-242: This is confusing and messy. Please, clarify and re-phrase.

Lines 243-245: Remove “because there was a small difference in yield among treatments, but there was a significant difference in irrigation water quantity. In addition, all the treatment areas in 2017 experienced a yield reduction”.

Line 247-248: Remove this last sentence.

Line 249: The title of this table does not stand alone and must be re-phrased.

Table 3: Check significance letters for plant height since the differences among treatments are not clear due to the great variability in the data. Besides, consider using just one decimal value. In 1000-grain weight there are no differences among treatments, why using “a”? Why F-value and not p-value? If you used p-values, asterisks and “ns” would not be needed. In the footnote, “Values within each column”, are you comparing the two years and treatments or only treatments within a given year? Please, clarify.

Discussion:

This section is very weak and not convincing. Many portions look like a repetition of results and they are badly written. Besides, the implications for sustainability (long-term irrigation, groundwater pollution due to fertilizer application and brackish water use, how the diluted brackish water for irrigation was obtained, quality of grains, etc.) are not discussed or even mentioned.

Lines 253-254: Remove “study” and “to provide water for crop growth”.

Lines 254-257: This is an introduction but not a discussion.

Lines 258-259: “it is difficult to implement these methods using the management methods of ordinary farmers”, this is messy and the meaning is unclear.

Lines 262-264: This sentence can be removed since it is not needed.

Lines 264: Remove “of field experiments”.

Lines 265-266: “On the premise of conserving the water supply of the wheat”, unclear meaning. Check English and re-phrase.

Line 268: “crop traits” instead of “agronomic traits of crops”.

Line 270: “amount” instead of “quota”.

Line 271: “crop growth” instead of “the growth of the crop”. What do you mean by “have obvious change”?

Line 274: “had less” instead of “solution grew the worst, with”. Remove “the least”.

Line 275: Remove “solution adopting brackish water irrigation” and “however”.

Lines 276-277: A citation is needed here.

Lines 278-281: Condense and improve English.

Line 282: Remove “Leaf is an important organ to maintain water balance of crop”.

Line 283: “crop stress” instead of “the stress on crop”.

Line 285: “leaf aging” instead of “aging of leaves”.

Line 286: “”T3 may be larger than”, why? Did you find any difference?

Line 287: “reduce the final yield” instead of “thus cut down the final output”.

Lines 287-288: Remove the last sentence since this is already knew.

Line 289: “Irrigation with brackish water has different effects on crop growth depending on its developmental stage” instead of “In different growth periods of crop, brackish water irrigation has different effects on crop growth”.

Line 291: Remove “we found”.

Lines 293-294: Remove “produced a growing influence on the photosynthesis of crop, and”, and “Then”.

Line 295: “stage” instead of “period”.

Lines 296-297: Remove “Before the filling period came, the photosynthesis rates of plants handled with various water irrigation modes didn’t have obvious difference”.

Line 298: “does not” instead of “doesn’t”. Remove “singly”. “Ors and Suarez” instead of “Ors et al.”

Line 299: Remove “research”. Use “yield” instead of “output” and “plant growth” instead of “the growth of plants”.

Lines 300-304: Too long and badly written. Please, re-phrase.

Line 305: “As in the study by” instead of “In the aspect of winter wheat output, our experiment gained the same results as”. Besides, it is “Jiang et al.” and not “Jiang et al.’s”.

Line 306: Remove “that is”. “increase of salt in irrigation water”; however, you did not increase salt concentration in irrigation water.

Lines 307-309: This sentence does not make any sense.

Line 309: “Zhang et al. [32] indicate” instead of “Zhang X. et al.’s [32] research indicates”.

Line 310: “reducing yield” instead of “so as to affect the final output of the crop”.

Line 312: Remove “From the outputs, we can see”.

Line 313: “the final yield of T3 was higher than those of T2 and T4, which did not received water during the filling stage” instead of “the final output of T3 solution is higher than those of T2 and T4 solutions not adopting irrigation”.

Lines 315-316: “while recognizing that the effect of brackish water on crops accumulates year by year”, what do you mean?

Line 317: “water” instead of “irrigation”. Remove “of the winter wheat”.

Line 318: “affecting photosynthesis, and reducing plant height” instead of “affecting the photosynthesis of the crops, and influencing plant height”.

Line 322: Remove “stress” after “drought”.

Line 323: Remove “the” before “final yield”.

Line 326: “Feng et al. [35], the desalination” instead of “Feng et al.’s [35] research, the desalination”.

Lines 332-336: This is not discussion but Results. What do you mean by “profit and loss”?

Line 337: “the irrigation water is more critical to soil salinity changes”, it depends on salt concentration.

Line 339: “that brackish water has a leaching effect on soil salinity”. Unclear meaning.

Lines 340-345: These ideas are not well described and should be further analysed and discussed.

Figure 10: This belongs to the Results section. Besides, here you combined a couple of tables with a couple of graphs, repeating information. Moreover, the symbols in the graph on the right side are not the same as those in the legend and in the other graph.

Line 347: The caption for this figure does not stand alone. Re-phrase it.

Conclusions:

Instead of bullet points, this section should be a single paragraph. They seem like a repetition of results. Authors should be more cautious.

Line 348: “Conclusions” instead of “Conclusion”.

Lines 349-351: Too long and English must be checked and revised.

Line 352: Check superscripts in the units. “can be used for irrigation in arid and semi-arid areas”, you should be more cautious because you only studied one crop species on one region.

Line 354: “the change of soil salinity still needs further observation”, re-phrase this.

Line 355: Remove “Our observations of the growth of winter wheat for two years indicate the following:”

Line 358: Check superscripts in the units.

Lines 360-361: Check superscripts in the units.

Line 361: “feasible irrigation method in arid and semi-arid areas”, not true. Only in your area, if you are optimistic because you did not discussed on the sustainability of this practice. Besides, in other areas, groundwater is not available. Be more prudent with your conclusions.

Line 363: “guarantee crop yield”, to a given extent.

References:

Please, check the reference list according the journal guidelines.

Line 371: “References” instead of “Reference”.

Line 418: “Arabidopsis thaliana” must be written in italics.

Line 428: “Yellow River Delta” instead of “yellow river delta”.

Lines 433-434: “Arabidopsis thaliana” and “Thellungiella salsuginea” must be written in italics.

Line 453: “Amaranthus” must be written in italics.

Line 456: “Capsicum annuum” must be written in italics.

Line 472: “Zea mays” must be written in italics.

Maybe you can use the following references for improving your introduction and discussion sections, since they are highly related to your study:

Letey, J.; Hoffmann, G.J.; Hopmans, J.W.; Grattan, S.R.; Suarez, D.; Corwin, D.L.; Oster, J.D.; Wu, L.; Amrhein, C. Evaluation of soil salinity leaching requirement guidelines. Agric. Water Manage. 2011, 98, 502-506; doi: 10.1016/j.agwat.2010.08.009

Mirás-Avalos, J.M.; Intrigliolo, D.S. Grape composition under abiotic constrains: Water stress and salinity. Front. Plant Sci. 2017, 8, 851; doi: 10.3389/fpls.2017.00851

Yuan, C.; Feng, S.; Huo, Z.; Ji, Q. Effects of deficit irrigation with saline water on soil water-salt distribution and water use efficiency of maize for seed production in arid Northwest China. Agric. Water Manage. 2019, 212, 424-432; doi: 10.1016/j.agwat.2018.09.019  

Author Response

First of all, thank you very much for your valuable comments on our manuscript. We are aware of our shortcomings in experiment and writing and your comments have greatly helped us to improve our manuscript. According to the editor's suggestion, I used the "Track Changes" to revise the manuscript. We used Microsoft Office Word's "Comments" function to explain your problem,  you can review these contents through the manuscripts I submit.

About your comments to our writing problems, we have corrected all these places. Therefore, in the manuscript, we do not respond to these comments one by one.

About your comments on our other issues, we also revised and explained them in our manuscript using Word's “Comments” function. 

We appreciate for your warm work earnestly and did our best to revise the manuscript, and hope that the correction will meet with approval. If you still have questions about our manuscript, we are still will to continue to revise our manuscript.

Once again, thank you very much for your comments and suggestions.

Round 2

Reviewer 2 Report

The authors made changes and additions and so the quality of the manuscript was improved.

However, at the address below we found an abstract of a similar paper, containing the experimental data obtained between 2015-2016. And in the manuscript are presented the data obtained during 2015-2017.

http://apps.webofknowledge.com/full_record.do?product=UA&search_mode=GeneralSearch&qid=11&SID=C24BtCuamaHTpD2VcYt&page=1&doc=1

The published work is not cited in the manuscript, and only Mr. Guibin Pang is a common author.

How can the authors explain this situation?

Reviewer 3 Report

The revised version of the manuscript entitled “Effects of Irrigating with Brackish Water on Soil Moisture, Soil Salinity, and Agronomic Response of Winter Wheat in Yellow River Delta” (Reference number Sustainability-49553-v2), authored by T. Wang, Z. Xu and G. Pang represents a great improvement from the previous version, which I had the opportunity to review. I appreciate the effort made by the authors and thank them for taking into account some of my comments and suggestions, although others essential for the better understanding of their research have been disregarded.

In fact, the Materials and Methods section still needs further information to be clear enough for allowing a replication of the experiment. Besides, I detected inconsistencies between text and figures in the Results section, as well as problems with English. The Discussion section is very weak.

Therefore, I recommend a major revision of this manuscript because, in its present form, it does not achieve the high-quality standards for being published in Sustainability.

In the following lines, I provide the authors with a number of suggestions in order to improve their work. Those comments refer to the PDF version of the manuscript, with the changes marked.

Specific comments to the authors:

Please, improve English by sending your manuscript to the editing support services of the journal or to a native English speaker. I have made a number of suggestions but my comments are not comprehensive.

Abstract:

Line 12: Remove “scientifically and”. Use “A two-year” instead of “A two years”.

Line 20: “irrigating with brackish water” instead of “irrigating brackish water”. Use “while” instead of “cwhile”.

Line 22: Remove “the result of T3 on soil salinity accumulation was very similar to T1”. It is not true since salt accumulated under T3 was 25% greater than that accumulated under T1.

Line 25: “greater” instead of “better”.

Line 26: “followed the rank order” instead of “followed the order of”.

Lines 27-28: Check English and re-phrase.

Introduction:

Line 40: “main” instead of “mainly”. Include “and” before “cannot”.

Lines 47-54: English must be greatly improved here. For instance, “to some extent” must be moved to after “with saline waters”. “EC” must be defined when first used. The lines 51-52 do not make sense since authors used different water qualities for two different crops and mixed the results. Definitely, these lines that have been added to this revised version of the manuscript need re-writing.

Line 62: “are carried out” instead of “were carried out”.

Lines 62-63: This sentence does not make sense. Please, re-phrase it.

Line 67: Remove “for irrigation”. Remove “irrigation” before “can alter the soil environment.

Line 72: “China” instead of “china”.

Lines 73-76: Re-phrase to “In the Yellow River Delta, flood irrigation is currently the main system used by winter wheat farmers. In order to avoid soil salinization when irrigating with brackish water, water quality and amount should be carefully controlled.”

Line 80: “for applying” instead of “of irrigating with”.

Materials and Methods:

This section needs further information to fully describe the experiment which has been carried out. What was the electrical conductivity of the fresh water? It is essential to know this in order to understand the observed effects and the differences among irrigation treatments.

Lines 91-92: “being overexploited in recent years” instead of “with the over exploitation of groundwater in recent years”.

Lines 103-105: This title for Table 1 is not correct. Authors should indicate what they are displaying on the table and do not state other soil characteristics.

Line 119: “such as” instead of “such like”.

Lines 123-124: Please, separate “one” and “at”.

Table 2: “Seasons” is not correct, it does not make sense. It should be “Dates of sowing and harvest and length of the growing cycle”.

Lines 129-132: How many measurements per treatment?

Lines 133-134: How many samples per replication?

Lines 136-138: These two sentences do not make sense. Please, re-phrase them. In fact, the way in which you obtained soil salinity in mass is not clear.

Line 141: “Crop” instead of “Crops’”.

Lines 142-147: This is rather confusing. I suggest authors to re-phrase it and condense the information.

Line 150: What do you mean by “sampled 10 cm on the ground”? What did you sampled?

Lines 152-153: This does not make sense.

Line 155: “dried” instead of “drying”. What do you mean by “to inactivation”?

Lines 164-165: This is rather weird. If you used ANOVA for analysing your data, why did you use the F-test for assessing the significance of the treatment effects? ANOVA already provides an assessment of the significance of treatment effects.

Results:

Lines 168-174: Authors must verify figure 5 since it seems that the data displayed on this figure are different from what they state in these lines. They said that no differences were not detected among water qualities; however, it seems that T2 and T4 at 0-20 cm were different in 2017 (Figure 5b), and the same should be said for T1 and T3 in 2016 (Figure 5a). More differences on soil water content among water qualities can be seen in 2017.

Line 182: Here, authors stated that the salt contents in the soil were similar for all treatments; however, when looking at figure 6, we can realize that T2 on the soil surface has twice salt more than T1 and T4.

Lines 184-198: This portion of text should be better described. I suggest condensing information and just stating the main results from the figure.

Line 208: Remove “whereas”.

Line 211: “indices” instead of “indexes”.

Line 225: “Daily evolution of photosynthesis rate” instead of “Physiological indexes”.

Lines 231-233: These sentences must be re-phrased for the sake of clarity.

Line 245: “the four treatment areas were basically the same”, this does not make sense. Re-phrase it.

Line 248: “greater” instead of “stronger”.

Line 251: “jointing-flowering”? Are not these graphs referred to the filling stage?

Lines 255-261: This paragraph must be re-phrased and simplified. It is easier to read, for instance, “Plant height was higher in T1 and lower in T2 for the both studied years” than what you have written. Therefore, improve English.

Line 261: Not only different when compared to T3 but also when compared to T4.

Line 272: Include “observed in” after “dry weight was”. Remove “had the worst dry matter results”.

Lines 275-283: It is interesting to note that a 33% reduction in irrigation water (independently of its quality) only reduced wheat yield by less than 19%. In this regard, it would be interesting to discuss why not reducing water amounts from the 240 mm that you consider common and combine this with the use of brackish water. For instance, what would occur if you use 200 mm (150 mm fresh and 50 mm brackish water)?

Table 3: Are you showing standard errors or standard deviations? Significant differences in plant height are not clear as you indicate in the table due to the variability in the data. The same could be said for block yield. Please, check the significances.

Discussion:

This section did not improve from the previous version of the manuscript. In fact, it repeats the results and do not provide an overview on the sustainability of the irrigation practice (long-term irrigation, groundwater pollution due to fertilizer application and brackish water use, how the diluted brackish water for irrigation was obtained, quality of grains, etc.) experimented here. This section is very weak and not convincing.

Lines 301-302: Not sure about these percentages. You used fresh water to dilute the brackish water so the water savings should be less than those percentages show in here. Please, check.

Lines 304-306: These sentences can be removed because they do not add anything new to the discussion.

Lines 308-309: This means that growth in dry matter is the same for all treatments, indicating that the quality and amount of irrigation affected only water accumulation in plant tissues. I suggest re-phrasing this sentence accounting for this issue.

Line 310: “T4 plants were shorter and with lower dry matter contents, while T3 plants were less affected” instead of “all plants of T4 had less the lowest heights and dry matters; the plant heights and dry matters of T3 were less influenced”.

Lines 311-314: This sentence is messy and too long. Revise it and enhance English.

Lines 316-317: Re-phrase this sentence to “Therefore, the effect of irrigation water amount on crop agronomic traits is more obvious than that of water quality”.

Line 323: This does not make sense. What is the difference between drought and water stress? LAI was less affected when compared with what?

Lines 332-338: This is messy and not clear. The explanation is not enough for discussing your findings.

Lines 339-346: This paragraph is badly written. Some of the comments must be given earlier in the manuscript, such as the choice of the number of irrigation events that must be stated in the Materials and Methods.

Lines 354-359: I do not agree with this. You only performed the study for a couple of years, it is impossible to extrapolate your results for a longer period, so you cannot conclude that using brackish water for irrigation is a feasible practice.

Lines 360-367: Improve English and the explanations on the similarities and differences between your study and previous research.

Lines 368-372: These are results, not discussion.

Figure 12 must be in results and not in the discussion section. Besides it includes a table that repeats information with respect to the graphs.

Conclusions:

From my viewpoint, this section provides a conclusion that has not been properly discussed in the former section. Moreover, English needs polishing.

References:

Maybe you can use the following references for improving your introduction and discussion sections, since they are highly related to your study:

Letey, J.; Hoffmann, G.J.; Hopmans, J.W.; Grattan, S.R.; Suarez, D.; Corwin, D.L.; Oster, J.D.; Wu, L.; Amrhein, C. Evaluation of soil salinity leaching requirement guidelines. Agric. Water Manage. 2011, 98, 502-506; doi: 10.1016/j.agwat.2010.08.009

Mirás-Avalos, J.M.; Intrigliolo, D.S. Grape composition under abiotic constrains: Water stress and salinity. Front. Plant Sci. 2017, 8, 851; doi: 10.3389/fpls.2017.00851

Yuan, C.; Feng, S.; Huo, Z.; Ji, Q. Effects of deficit irrigation with saline water on soil water-salt distribution and water use efficiency of maize for seed production in arid Northwest China. Agric. Water Manage. 2019, 212, 424-432; doi: 10.1016/j.agwat.2018.09.019